# Prevalence and Risk Factors of *Eimeria* spp. in Broiler Chickens from Pichincha and Santo Domingo de los Tsáchilas, Ecuador

**DOI:** 10.3390/pathogens13010048

**Published:** 2024-01-04

**Authors:** Ana Cevallos-Gordon, C. Alfonso Molina, Nilda Radman, Lenin Ron, María Ines Gamboa

**Affiliations:** 1Faculty of Veterinary Medicine, Central University of Ecuador, Quito 170521, Ecuador; camolina@uce.edu.ec (C.A.M.); ljron@uce.edu.ec (L.R.); 2Faculty of Veterinary Sciences, The National University of La Plata, La Plata B1900AFW, Argentina; nildarad@yahoo.com.ar; 3Institute of Public Health and Zoonotic Research, Central University of Ecuador, Quito 170521, Ecuador; 4Faculty of Agronomy, Central University of Ecuador, Quito 170521, Ecuador

**Keywords:** coccidiosis, *Eimeria*, OPG (oocysts per gram), risk factors, poultry farming, Ecuador

## Abstract

Coccidiosis in chickens is a parasitic disease of economic importance for the poultry industry. In Ecuador, there is limited information regarding the prevalence of *Eimeria* spp. on commercial broiler farms. Therefore, a total of 155 poultry farms in the provinces of Pichincha and Santo Domingo de los Tsáchilas were surveyed. The analysis of fresh fecal samples was conducted to determine the parasitic load of six of the seven chicken *Eimeria* species (excluding *E. mitis*) through multiplex PCR. Additionally, an epidemiological survey was performed to assess the risk factors associated with the infection using a multivariable logistic regression model. All samples tested positive for the presence of *Eimeria* spp., despite the farmers having implemented prophylactic measures, and no clinical coccidiosis cases were recorded. The parasitic load varied between 25 and 69,900 oocyst per gram. The species prevalence was as follows: *Eimeria* spp. 100%, *E. maxima* 80.4%, *E. acervulina* 70.6%, *E. praecox* 55.4%, *E. tenella* 53.6%, *E. necatrix* 52.2%, and *E. brunetti* 30.8%. The main species combination was *E. cervuline*, *E. maxima*, *E. necatrix*, and *E. praecox* (23.90%), followed by *E. tenella*, as a unique species (10.69%), and then E. *acervulina*, *E. maxima*, and *E. praecox* (8.81%). It was observed that farms operated by independent producers had a higher amount of *Eimeria* spp. and higher probability of the presence of *E. brunetti*, *E. necatrix*, *E. praecox*, and *E. tenella*. Poultry houses located below 1300 m above sea level were associated with a higher parasitic load and the presence of *E. brunetti*. Birds younger than 35 days of age and from open-sided poultry houses (with rudimentary environmental control) had a higher probability of presenting *E. maxima*. Drinking water from wells increased the risk of *E. praecox* presence. Research aimed at designing control strategies to improve health management on poultry farms in the region would help minimize the impact of coccidiosis.

## 1. Introduction

Avian coccidiosis is a parasitic disease of significant importance in broiler chickens [1,2,3,4,5,6,7] with a high economic impact on the global poultry industry [8,9,10,11,12,13]. Several factors contribute to its persistence, such as the parasite’s direct life cycle, lack of cross-protection in poultry houses, and ideal environmental conditions for oocyst sporulation [7]. Furthermore, it has been reported that avian coccidiosis predisposes birds to bacterial diseases of the intestine, particularly necrotic enteritis caused by *Clostridium perfringens*, a condition that also results in significant economic losses and mortality in birds [14]. Moreover, as the bird density increases, the risk of coccidiosis infection also rises [15]. Poultry industrialization, and its trend to increase bird density in poultry houses, as well as the reduction in sanitation downtime between stocks, contribute to the rise in oocyst concentration in the litter material and lead to a higher risk of parasitic infection [16].

In 1984, according to its incidence and importance in the poultry industry of Central and South America, avian coccidiosis was reported as the fourth most relevant disease in this part of the continent [17]. However, in Latin America, precise information regarding the prevalence, incidence, and impact on the poultry industry is not available [18,19]. In Ecuador, there are few reports on the presence of coccidiosis in commercial birds [20,21,22], and to date, its prevalence and economic losses in domestic birds have not been determined.

The prophylactic use of anticoccidials in feed has had a profound impact on the poultry industry [23,24]. Nevertheless, due to the development of resistance to anticoccidials, current management programs have been inadequate in controlling this disease [25,26]. Additional strategies, such as increasing biosecurity measures, implementing good management practices and vaccination, have not been sufficient in coccidiosis control [17].

The identification of *Eimeria* species provides a foundation for the prevention, surveillance, and control of coccidiosis [27,28]. Determining the parasitic load and species composition on each farm should be considered fundamental regarding decision-making about the implementation of anticoccidial and vaccination programs [29]. Furthermore, several environmental and management factors have been associated with the presence and severity of coccidiosis, among which include bird age [30,31,32], stocks size [30], geographical location, litter depth [30], poor sanitation conditions [32], and poor environmental management [33]. On the other hand, the widespread use of drugs has exacerbated *Eimeria* resistance to anticoccidials [34], becoming a huge problem in the poultry industry [2,13,35]. However, it has also been observed that high hygiene standards and strict biosecurity programs are associated with lower losses [36].

Currently, it has been estimated that worldwide losses for the industry due to coccidiosis amount to between 10 and 14 billion dollars [37]. These values include expenses related to prophylactic and therapeutic measures, as well as production losses [38,39], making coccidiosis an ongoing major challenge faced by the global poultry industry [9,15]. For all these reasons, an analysis of the prevalence of *Eimeria* spp. in the Ecuadorian poultry industry, as well as sanitary factors that could determine its presence and variations in parasite load, promised to be beneficial for both current and future practices.

## 2. Materials and Methods

### 2.1. Ethics Statement

The project was approved by the Institutional Committee for the Care and Use of Laboratory Animals of Faculty of Veterinary Sciences of the National University of La Plata [91-2-19T], and research procedures were carried out in accordance with national and institutional regulations.

### 2.2. Study Area and Farms

The research was conducted in the north-central region of Ecuador, specifically in the provinces of Pichincha and Santo Domingo de los Tsáchilas, which are located in the foothills and western flanks of the Andes Mountain range and are the major broiler-chicken production areas [40] (see Figure 1).

### 2.3. Sampling and Sample Analysis

The sampling was directed at commercial broiler-chicken farms in their final rearing phase. The poultry census of 2015 [41] was used as the reference frame for the sampling, determining a sample size of 155 farms. Whereas a total of 159 fresh feces samples were obtained on 155 farms, a double sampling was conducted due to the temporal difference between production stocks.

The birds in the sample were in the last week of the growth period. For sample collection, the method described by Kumar et al. (2014) [42] was used. This included tracing a W-shaped pattern along each poultry house to collect fresh feces [regular and cecal feces]. Between 15 and 200 samples were collected from each poultry house. The samples were transported in sealed bags at 4 °C and processed at the Central University of Ecuador Laboratories of the Institute of Research in Zoonoses at the (CIZ). Each sample was homogenized, and 20 g of feces were reserved for the analyses. Then, the Willis flotation technique was employed to confirm the presence of oocysts. Subsequently, the parasitic load (OPG, oocyst per gram of feces) was quantified using the modified McMaster technique [43]. Observations were made using a 10× objective (Carl Zeiss—Axiostar plus microscope) for oocyst counting. DNA extraction was performed using the commercial FastDNA Spin Kit for Feces [MP] following the manufacturer’s instructions. Commercial vaccines Fortegra^®^ (*E. acervulina*, *E. maxima*, *E. maxima* MFP, *E mivati*, and *E. tenella*) and Coccivac D2^®^ (*E. acervulina*, *E. maxima*, *E mivati. E. brunetti*, *E. necatrix*, and *E. tenella*) from MSD laboratory were used as positive controls. First, PCR was individually amplified for each species. Then, multiplex amplification was carried out, using the technique published by Fernandez et al. (2003) [44] as a reference; the primer sequences of *Eimeria* species can be observed in Table 1. The multiplex PCR was conducted on the BIOER thermocycler (C-1000 Touch). Amplification products were analyzed through electrophoresis, using a 2% agarose gel stained with ethidium bromide (Invitrogen™). In the case of *E. mitis*, PCR could not be standardized for this species due to the lack of positive control, so only the remaining six species were analyzed.

### 2.4. Farm Characteristics

All sampled farms, based on their characteristics and capacity, were of commercial type (Agrocalidad and CONAVE). Their location was georeferenced. The detailed variables investigated can be found in Appendix A.

### 2.5. Identification of Risk Factors

To identify the risk factors associated with parasitic load and the presence of different *Eimeria* species, an epidemiological “Epidemiological Survey” questionnaire was developed and validated by national and international experts. It consisted of observing the characteristics of the infrastructure of the production centers and conducting interviews with the head of each poultry farm. The information collected through the questionnaires was entered into a database using the Microsoft Excel^®^ program for organization and cleaning purposes.

### 2.6. Statistical Analysis

To compare the relationship between the characteristics of poultry farms and the control measures used for *Eimeria* spp. with the mean number of oocysts per gram of feces (OPG), analysis of variance (ANOVA) was employed. OPG data were log transformed (log(x + 1)). Shapiro–Wilk normality test for model residuals was also performed. For variables with three or more levels, the Tukey test was used to determine if there were significant differences. 

The prevalence of each *Eimeria* species (P) was calculated with a 95% confidence interval. When determining the prevalence of *E. acervulina*, *E. maxima* and *E. tenella*, samples from birds that received anticoccidial vaccines were not considered. This was due to the use of vaccines with live oocysts, which were likely to contain oocyst from vaccines rather than field species or strains [43].

To determine the level of association between the presence/absence of each *Eimeria* spp. species and the different analyzed factors, the Odds Ratio (OR) was used, while Fisher’s exact test was used to evaluate the association of each factor. Multiple logistic regression was used to identify the covariates for the final model, considering all factors with a *p*-value less than 0.20. Prior to the analysis, the Variance Inflation Factor (VIF) was considered to avoid collinearity between variables, discarding variables with a VIF level greater than 8. The stepwise procedure was used for the selection of the final model. The statistical analysis was performed using R statistical software [45] version 4.0.2. The level of statistical significance was set at 5% for the entire study. For map creation and georeferencing, the statistical package Mapview in the R environment was used [46].

## 3. Results

### 3.1. General Information and Farm Management

The 159 samples analyzed between June 2019 and August 2020 were positive for the presence of *Eimeria* spp. oocysts. The farms exhibited varying levels of production ranching from low-technological family farms to highly technical industrial farms, all in conventional confinement with different installation capacities, ranging from 2000 to 376,444 birds. A detailed description of the variables can be found in Appendix A. 

### 3.2. Parasitological Results

All evaluated farms presented *Eimeria* spp. oocysts, meaning a prevalence of 100%. The only parasitic genus identified in the birds was *Eimeria* spp. No symptoms of poultry coccidiosis were observed on any farm. The parasitic load ranged from 25 to 69,900 OPG. Through PCR, six of the seven chicken *Eimeria* species were identified: *E. acervulina*, *E. brunetti*, *E. maxima*, *E. necatrix*, *E. praecox*, and *E. tenella*. Figure 2 demonstrates representative gel images showing the different band sizes for the amplicons corresponding to each of different *Eimeria* spp. detected. Note that gel images have been trimmed. The prevalence of *Eimeria* spp. per farm is shown in Table 2.

The OPG levels of the analyzed samples did not present any association with the presences of the *Eimeria* spp. species identified by PCR (*p* > 0.05). The most common combination found was *E. acervulina*, *E. maxima*, *E. necatrix*, and *E. praecox*, present in 23.90% of the samples, followed by *E. tenella*, as a unique species, which was present in 10.69% of the samples. Next were *E. acervulina*, *E. maxima*, and *E. praecox* in 8.81% of the samples; *E. tenella* and *E. brunetti* in 6.92% and *E. acervulina*, *E. maxima*, and *E. necatrix* in 5.66%. The combinations of *Eimeria* species can be observed in Table 3, and the geographic location of the most frequent combinations in Figure 3.

### 3.3. Association between Risk Factors and Eimeria *spp.* Parasitic Load

The variables that showed association with higher parasitic loads were farms located at altitudes below 1300 Meters Above Mean Sea Level (m.a.s.l.), poultry houses with dirt floors, younger birds (less than 35 days old), and evidence of the use of purchased feed. A lower OPG level was observed in drinking water sources from potable water, followed by water from rivers or canals and wells, with the latter two groups indicating performance of water treatment (See Table 4). Further details of the results are shown in Appendix A.

The final multiple regression model for the risk of the quantity of *Eimeria* spp. oocysts (OPG) shows that independent poultry farms had a higher number of oocysts compared to integrated farms (*p* < 0.01), with values ranging from 5460 to 9051 oocyst per gram, respectively. Altitudes below 1300 m.a.s.l. showed a higher number of oocysts, OPG = 8338.06, compared to farms at higher altitudes, OPG = 2582.42 (See Table 4).

The remaining variables analyzed, including the protective measures used by the surveyed poultry farmers, did not present any association with the quantity of *Eimeria* spp. oocysts.

### 3.4. Association between Risk Factors and Presence of Distinct Species of de Eimeria *spp*.

The variables that showed a positive association with the presence of different *Eimeria* species (Table 5) were as follow: the type of producer, where the independent producer level exhibited a higher risk with an OR of 2.13 for *E. acervulina* (*p* < 0.05) compared to integrated producers, and OR = 2.84 for *E. brunetti* [*p* < 0.01]; OR = 2.321 for *E. necatrix* (*p* < 0.05); OR = 2.48 for *E*. *praecox* (*p* < 0.01); and OR = 3.22 for *E. tenella* (*p* < 0.01) when compared to integrated producers. Dirt floors presented an OR of 4.28 times higher risk for *E. acervulina* (*p* < 0.01), and 2.31 for *E. praecox* (*p* < 0.01), compared to farms with cement floors. Wooden structures had an OR of 2.63 times higher risk of presenting *E. acervulina* (*p* < 0.01) and OR = 2.44 for *E. praecox* (*p* < 0.01), compared to metal structures. The origin of drinking water, when sourced from a well [groundwater], showed an OR of 2.79 higher risk for *E. acervulina* (*p* < 0.01]; OR = 2.57 for *E. maxima* (*p* < 0.05); and OR = 2.77 for *E. praecox* (*p* < 0.01), compared to water from rivers or canals. Not producing their own feed, and having no control over anticoccidial management, presents an OR = 12.11 times higher risk for *E. acervulina* (*p* < 0.01); 4.73 for *E. brunetti* (*p* < 0.01); 6.94 for *E. maxima* (*p* < 0.01); 2.83 for *E. praecox* (*p* < 0.01]; and 7.48 for *E. tenella* (*p* < 0.01), indicating a greater likelihood of presence compared to those who manufacture themselves or have a long-term relationship with the feed supplier.

The presence of *Alphitobius diaperinus*, a specie of beetle, showed an OR = 2.31 times higher risk of presenting *E. maxima* (*p* < 0.05); and OR = 1.98 times higher risk for *E. tenella* (*p* < 0.05) compared to when the insect was not detected. Birds within the youngest age range [below 35 days] presented an OR = 6.73 higher risk of *E. acervulina* infection (*p* < 0.01); 4.69 for *E. maxima* (*p* < 0.01); 2.32 for *E. necatrix* (*p* < 0.01); and 3.79 for *E. praecox* (*p* < 0.01) compared to birds older than 46 days. Intermediate-aged birds (35 to 45 days) showed a 2.84 higher risk of presenting *E. necatrix* (*p* < 0.05) compared to older birds. Farms that experienced clinical coccidiosis during breeding and used anticoccidial treatments in drinking water showed an OR = 4.21 times higher risk of presenting *E. brunetti* (*p* < 0.01) and OR = 3.73 for *E. tenella* (*p* < 0.05). The use of hydrated lime in floor disinfection showed an OR = 3.12 higher risk of presenting *E. acervulina* (*p* < 0.01) and 2.41 of *E. maxima* (*p* < 0.05), compared to producers who abstained from use. The rest of the analyzed variables did not show statistical association. Further details of the results can be found in Appendix A.

The final multivariable binary logistic regression model (See Table 4) determined that independent poultry farms had a higher risk of presenting *E. brunetti* with an OR = 2.79 (*p* < 0.05); OR = 2.21 (*p* < 0.05) for *E. necatrix*; 3.28 (*p* < 0.01) for *E. praecox* and OR = 3.69 (*p* < 0.01) for *E. tenella*, compared to producers integrated into poultry companies. Birds raised at altitudes below 1300 m.a.s.l. showed OR = 5.94 (*p* < 0.01) times higher possibility of presenting *E. brunetti* and 3.43 (*p* < 0.01) for *E. tenella*, compared to farms situated at higher altitudes. Open. sided poultry houses showed a positive association of 3.98 (*p* < 0.05) with the presence of *E. maxima*, compared to controlled-environment poultry houses. Poultry houses with wooden structures (roof support) had a risk OR = 2.99 (*p* < 0.05) for the presence of *E. acervulina* and 3.47 (*p* < 0.01) for *E. praecox*, compared to poultry houses with metal structures. Drinking water from well sources showed a positive association of OR = 2.59 (*p* < 0.05) with the presence of *E. praecox* compared to water received from rivers or canals. In birds younger than 35 days, the risk increases by OR = 6.93 times (*p* <0.01) in the presence of *E. maxima.*

## 4. Discussion

This research evidenced the presence of *Eimeria* spp. in 100% of the samples from asymptomatic birds, even though the sampled poultry establishments maintained some level of biosecurity. *Eimeria* spp. was the only parasitic genus found, demonstrating that the biosecurity measures were not sufficient to control the presence of this protozoan in commercial poultry farms, indicating an endemic pattern.

In Ecuador, the presence of *Eimeria* spp. in commercial broiler chickens has been poorly studied. In this regard, Villareal monitored the digestive tract of chickens at processing plants and reported that 5% presented caeca lesions suggestive of *E*. *tenella*, and 13% had macroscopic lesions [enteritis] [21]. Furthermore, when evaluating litter recycling on a farm in the locality of Pedro Vicente Maldonado, Pichincha Province, it was observed that all the litter had different OPG of *Eimeria* spp. at the end of the cycle [20]. Poultry coccidiosis was also reported as affecting commercial birds in eight out of thirteen farms analyzed in the Galápagos Islands, Ecuador [22].

*Eimeria* spp. has been reported in commercial birds from various continents [38]. For instance, in Romania, a prevalence of 91% was reported [25], in China 97.17% [47], in the northern region of India 81.3% [11], while in Ethiopia, it was 42.2% [36]. In countries like Mexico, despite the use of rotation of anticoccidial drugs in the feed, *Eimeria* spp. was detected in 100% of the sampled farms [48]. Similarly, in the states of Tocantins [18] and Bahía [49] in Brazil, *Eimeria* spp. was also detected in 100% of the farms. In the state of Santa Catarina, the reported frequency was 96% [12]. In Argentina, a prevalence of 88.37% was recorded [50], and in Colombia, it was found to be 92.8% [19]. Moreover, even in controlled-environment poultry houses in Brazil, with negative pressure ventilation, a prevalence of 87.5% for *Eimeria* spp. was reported [3,51].

Although there is little published information about the disease in Ecuador, all producers are aware of its existence and the economic losses it entails. This may be the reason they maintain permanent prophylactic measures. For example, 97% of surveyed farms use permanent anticoccidial programs in the feed, while 3% of them manage anticoccidial vaccines to prevent economic losses, similar to the results reported in Colombia [19]. Several reasons could justify the limited use of anticoccidial vaccines, including the cost, which is often higher compared to anticoccidial drugs [52]. Another reason could be the lack of perception in improving production parameters with the use of vaccines [53]. On the other hand, the use of anticoccidial vaccines has been associated with the proliferation of *Clostridium* spp. and the occurrence of necrotic enteritis [53]. Additionally, improper management of litter moisture complicates the vaccine response, leading to the development of clinical coccidiosis [43].

The parasitic load of *Eimeria* spp. in broiler chickens in their last week of life varied between 25 and 69,900 OPG, with no clinical symptoms of coccidiosis in the birds. These results align with other research findings [13,19,54]. Different *Eimeria* species presented varying degrees of pathogenicity [55,56], leading to variations in the number of oocysts required to produce intestinal lesions [55]. Therefore, it is crucial to know the parasitic concentration and the species composition of *Eimeria* spp. [29]. Moreover, the parasitic presence can be influenced by several factors [57], including different levels of sensitivity to anticoccidial drugs [13]. The wide range in the parasitic concentration of *Eimeria* spp. is likely due to the use of anticoccidials, as various types of drugs and programs were identified. The specific anticoccidial programs used, withdrawal times, and rotation of active ingredients could not be established because the interviewees were either unaware or had inadequate information. This finding coincides with what has been reported in poultry producers in Colombia [19]. In Ecuador, regulations for the use of drugs in poultry production are not clearly designed to ensure their responsible use and to minimize the risk of resistance development.

Through PCR analysis, six species were identified: *E. acervulina*, *E. brunetti*, *E. maxima*, *E. necatrix*, *E. praecox*, and *E. tenella*. *E. mitis* was excluded from this research, making its existence uncertain. Further investigation must be carried out to establish the true prevalence of this poultry species. Each sample represented between one and six species. The prevalence of *E. acervulina* was 70.59%, for *E. maxima* 80.39%, and for *E. tenella* 53.59%. These species have also been reported in other studies [3,5,19,25,37,58,59,60]. These results align with previous research indicating that *E. acervulina*, *E. maxima*, and *E. tenella* are commonly detected species worldwide [51,61,62,63]. In this research, the most frequent species combination was *E. acervulina*, *E. maxima*, *E. necatrix*, and *E. praecox* with 23.90%, and then *E. tenella* as a unique species with 10.69%. Moreover, *E. tenella* and *E. necatrix* are considered the most pathogenic species [47,61,63]. The prevalence of *E. necatrix* was 52.20%, a higher value compared to other studies [6,19,47,59], while in other studies, some authors did not detect it [3,60]. The presence of *E. necatrix* is less likely in broiler chickens, but it is more common in older birds and in tropical regions [61]. This could justify our findings, as 77% of the evaluated farms were located at altitudes below 1300 m.a.s.l. The prevalence of *E. praecox* was 55.35%, and *E. brunetti* was 30.82%. These species have also been reported in other studies [6,25,60,64].

In the final model, the variables that showed an association with the presence of *Eimeria* spp. and their species were independent producers, birds raised at altitudes below 1300 m.a.s.l., poultry houses with wooden structures, birds younger than 35 days of age, open-sided poultry houses, and drinking water from wells. The variable ¨independent producers¨ showed a high association with greater amounts of OPG of *Eimeria* spp. and a higher probability of presenting *E. praecox*, followed by *E. brunetti*, *E. necatrix*, and *E. tenella*. These last three species of *Eimeria* have been linked to high pathogenicity in broiler chickens [65]. It is worth noting that this group of poultry farmers have established strict biosecurity systems and, due to their capacity and management, they are considered commercial-type poultry farmers [66]; furthermore, all of them use chemoprophylaxis. The presence of cross-resistance and multi-resistance to anticoccidials is quite common, which complicates the control of coccidiosis through drug rotation [2]. On the other hand, genetic variations in *Eimeria* species have been reported between the northern and southern hemispheres [67], and it is believed that this variation is attributable to the use of anticoccidial drugs and vaccines [15]. Additionally, this type of producer tends to have deficient management systems in poultry rearing, which have been associated with coccidiosis [27,36]. Unlike other studies, the size or capacity of the farm did not show any statistical association with parasite load or the prevalence of *Eimeria* [5,68]. Birds raised at altitudes below 1300 m.a.s.l. presented a higher risk of exhibiting higher OPG values and *Eimeria* spp., *E. brunetti*, and *E. tenella*. These results could be related to the typical environmental conditions [tropical and subtropical] of these altitude levels in Ecuador, such as higher prevalence of coccidiosis at altitudes below 4000 feet [1291 m.a.s.l.]. Altitude and associated temperature are natural environmental factors that affect parasite abundance [69]. The altitude levels of the studied areas in Ecuador could interfere with the sporulation of *Eimeria* oocysts. Lower altitude levels [subtropical zones] exhibit higher relative humidity and ambient temperature [68], conditions that could favor oocyst sporulation. The quantity of *Eimeria* oocysts is positively related to the level of humidity [70]. Similarly, higher ambient-relative humidity is associated with increased oocyst excretion [48,59].

However, it should be taken into consideration that in Ecuador, birds raised at higher altitudes undergo restriction or controlled feeding to reduce metabolic diseases [71]. This practice results in the birds reaching their market weight at a later age. It has been reported that young chickens are more vulnerable to coccidiosis [72]. The younger age of the birds (less than 35 days) was identified as a risk factor for the presence of *E. maxima* and highest parasitic load of *Eimeria* oocyst. Several studies agree with these findings [36,58,59,72].

Open-sided poultry sheds showed a higher risk of exhibiting *E. maxima* compared to controlled-environment sheds. These results align with previous research, which suggests that improving environmental control enhances coccidiosis control [3,15,36]. Inadequate environmental management in poultry sheds has been reported to provide ideal conditions for oocyst sporulation [7]. Poultry sheds with wooden structures also showed a higher risk of presenting *E. acervulina* and *E. praecox*. The wooden material in contact with poultry litter is likely an important reservoir for oocysts. Sporulated oocysts are highly resistant to environmental conditions [8,16,73]. In this context, the United States Department of Agriculture in Beltsville, Maryland, reported oocyst survival in the ground for 602 days [55]. Moreover, wooden structures tend to be associated with dirt floors in poultry production in Ecuador.

The “well water” (groundwater) factor showed a higher risk for *E. brunetti* and *E. praecox* presence. This leads to a presumption that groundwater may be contaminated, and the purification methods used might not effectively eliminate *Eimeria* oocyst.

## 5. Conclusions

Poultry coccidiosis is affecting the central-northern region of Ecuador. *Eimeria* spp. was present in 100% of the farms, with the presence of a least six species. Certain types of producers, farms at altitudes below 1300 m.a.s.l., wooden infrastructure, birds younger than 35 days, open-sided poultry sheds, and well water for drinking showed a higher risk of *Eimeria* species presence. Research aimed at designing control strategies to improve sanitary management on poultry farms in the region would help minimize the impact of coccidiosis.

## Figures and Tables

**Figure 1 pathogens-13-00048-f001:**
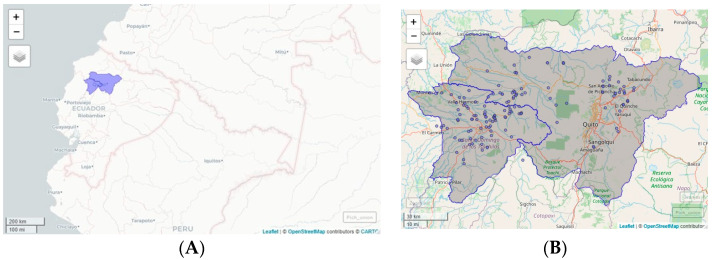
(**A**) Map of Ecuador. Surveyed area: Pichincha y Santo Domingo de los Tsáchilas. (**B**) Farm poultry distribution.

**Figure 2 pathogens-13-00048-f002:**
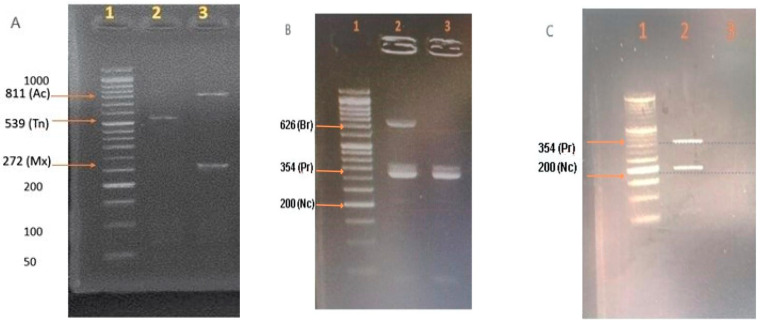
Identification of *E. acervulina* (Ac), *E. brunetti* (Br), *E. maxima* (Mx), *E. necatrix* (Nc), *E. praecox* (Pr), and *E. tenella* (Tn) using PCR. Agarose gel electrophoresis (2%) stained with ethidium bromide (Invitrogen™) was performed, and the results were observed as follows: Molecular weight marker 1 (**A**–**C**): *E. acervulina* (811 bp): samples 3 (**A**); *E. brunetti* (626 bp): sample 2 (**B**); *E. maxima* (272 bp): samples 3 (**A**); *E. necatrix* (200 bp): sample 2 (**C**); *E. praecox* (354 bp): samples 2 and 3 (**B**) and sample 2 (**C**); and *E. tenella* (539 bp): samples 2 (**A**). The DNA marker is G016 (100 bp Opti-DNA Marker (50–1500 bp) abm^®^).

**Figure 3 pathogens-13-00048-f003:**
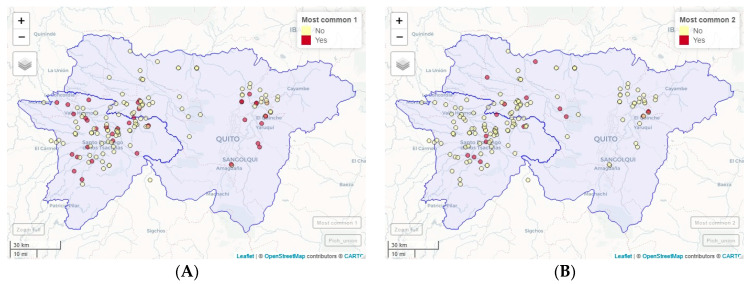
Geographic location of the most frequent combinations of *Eimeria* species. (**A**) Most common 1: *E. acervulina*, *E. maxima*, *E. necatrix*, and *E. praecox*. (**B**) Most common 2: *E. tenella*, (as a unique species). (**C**) Most common 3: *E. acervulina*, *E. maxima*, and *E. praecox*. (**D**) Most common 4: *E. tenella* and *E. brunetti*.

**Table 1 pathogens-13-00048-t001:** Primer sequences of *Eimeria* spp. from domestic birds for the multiplex PCR assay [44].

Species	SCAR	Primer Identification	Primer Sequence	Amplicon Size (pb)
*E. acervulina*	Ac-A03-811	Ac-01-F	AGTCAGCCACACAATAATGGCAAACATG	811
		Ac-01-R	AGTCAGCCACAGCGAAAGACGTATGTG	
*E. brunetti*	Br-J18-626	Br-01-F	TGGTCGCAGAACCTACAGGGCTGT	626
		Br-01-R	TGGTCGCAGACGTATATTAGGGGTCTG	
*E. tenella*	Tn-K04-539	Tn-01-F	CCGCCCAAACCAGGTGTCACG	539
		Tn-01-R	CCGCCCAAACATGCAAGATGGC	
*E. praecox*	Pr-A03-718	Pr-01-F	AGTCAGCCACCACCAAATAGAACCTTGG	354
		Pr-01-R	GCCTGCTTACTACAAACTTGCAAGCCCT	
*E. maxima*	Mx-A09-1008	Mx-01-F	GGGTAACGCCAACTGCCGGGTATG	272
		Mx-01-R	AGCAAACCGTAAAGGCCGAAGTCCTAGA	
*E. necatrix*	Nc-M02-1081	Nc-01-F	TTCATTTCGCTTAACAATATTTGGCCTCA	200
		Nc-01-R	ACAACGCCTCATAACCCCAAGAAATTTTG	

**Table 2 pathogens-13-00048-t002:** Prevalence of six of the seven poultry *Eimeria* species in the population of broiler chickens from the provinces of Pichincha and Sto. Domingo de los Tsáchilas.

	Samples Analyzed	Positive Samples	Prevalence [95% CI]
*E. acervulina*	153 *	108	70.59 (62.59–77.53)
*E. brunetti*	159	49	30.82 (23.88–38.70)
*E. maxima*	153 *	123	80.39 (73.03–86.19)
*E. necatrix*	159	83	52.20 (44.17–60.13)
*E. praecox*	159	88	55.35 (47.27–63.16)
*E. tenella*	153 *	82	53.59 (45.38–61.63)

* *E. acervulina*, *E. maxima* y *E. tenella* were not considered in the farms that received vaccines, as it is assumed that these species are present.

**Table 3 pathogens-13-00048-t003:** Association of different species of *Eimeria* in broiler-chicken farms from the provinces of Pichincha and Sto. Domingo de los Tsáchilas.

*E. acervulina*	*E. maxima*	*E. tenella*	*E. brunetti*	*E. necatrix*	*E. praecox*	Samples Positives (N 159)	Proportion (%)
+	+	+	+	+	+	4	2.52
+	+	+	+	+	-	3	1.89
+	+	+	+	-	+	4	2.52
+	+	+	+	-	-	2	1.26
+	+	+	-	+	+	7	4.40
+	+	+	-	+	-	1	0.63
+	+	+	-	-	+	4	2.52
+	+	+	-	-	-	1	0.63
+	+	-	+	+	-	1	0.63
+	+	-	+	-	+	1	0.63
+	+	-	+	-	-	3	1.89
+	+	-	-	+	+	38	23.90
+	+	-	-	+	-	9	5.66
+	+	-	-	-	+	14	8.81
+	+	-	-	-	-	6	3.77
+	-	+	+	-	-	1	0.63
-	+	+	+	+	+	2	1.26
-	+	+	+	+	-	1	0.63
-	+	+	+	-	-	4	2.52
-	+	+	-	+	+	1	0.63
-	+	+	-	+	-	3	1.89
-	+	+	-	-	-	1	0.63
-	+	-	+	+	+	1	0.63
-	+	-	-	+	-	2	1.26
-	+	-	-	-	-	2	1.26
-	-	+	+	+	+	1	0.63
-	-	+	+	-	+	3	1.89
-	-	+	+	-	-	11	6.92
-	-	+	-	-	+	1	0.63
-	-	+	-	-	-	17	10.69

**Table 4 pathogens-13-00048-t004:** Association between risk factors and the presence of *Eimeria* spp. and its species in the population of broilers chickens in the provinces Pichincha and Sto. Domingo de los Tsáchilas.

	OPG		*E. acervulina*		*E. brunetti*		*E. maxima*		*E. necatrix*		*E. praecox*		*E. tenella*	
Parameter	Media	*p*-Value	OR (95% CI)	*p*-Value	OR (95% CI)	*p*-Value	OR (95% CI)	*p*-Value	OR (95% CI)	*p*-Value	OR (95% CI)	*p*-Value	OR (95% CI)	*p*-Value
DEPENDENCE														
Integrated			Reference			Reference			Reference		Reference		Reference	
Independent			2.13 (1.00–4.72)	0.04 *	2.84 (1.35–6.08)	0.01 **			2.21 (1.11–4.47)	0.02 *	2.48 (0.23–5.10)	0.01 **	3.22 (1.59–6.67)	0.01 **
ALTITUDE														
Below 1300MAMSL	8338.06	0.00 **												
Above 1300MAMSL	2582.43													
FLOOR MATERIAL														
Cement	2614.1	0.00 **	Reference								Reference			
Ground	11,127.22		4.28 (1.98–9.67)	0.00 **							2.31 (1.17–4.6)]	0.01 **		
ROOF SUPPORT MATERIAL														
Metal			Reference								Reference			
Wood			2.63 (1.26–5.56)	0.01 **							2.44 (1.21–4.97)	0.01 **		
WATER SOURCE														
River or canal	3506.05 ^ab^	0.045 *	Reference				Reference				Reference			
Well	8734.63 ^a^		2.79 (1.22–6.46)	0.01 **			2.57 (1.04–6.33)	0.03 *			2.77 (1.23–6.41)	0.01 **		
Potable water	2603.85 ^b^		1.51 (0.36–6.97)	0.75			1.85 (0.38–12.17)	0.51			1 (0.21–4.27)	1		
ANTICOCCIDIAL MANAGEMENT THROUGH FOOD														
Yes	4939.46	0.00 **	Reference			Reference	Reference				Reference		Reference	
No	13,356.05		12.11 (2.89–108.61)	0.00 **	4.73 (2.06–11.16)	0.00 **	6.94 (0.64–63.08)	0.00 **			2.83 (1.21–7.12)	0.01 **	7.48 (2.8–23.5)	0.00 **
*Alphitubius diaperinus*														
No							Reference						Reference	
Yes							2.31 (0.02–5.32)	0.03 *					1.98 (1.00–4.00)	0.01 *
AGE														
More than 46 days	3021 ^b^	0.01 **	Reference				Reference		Reference		Reference			
Less than 35 days	12,388 ^a^		6.73 (2.32–22.64)	0.00 **			4.69 (0.48–17.84)	0.01 **	2.32 (1.00–5.51)	0.01 **	3.79 (1.58–9.41)	0.00 **		
Between 36 and 45 days	5266 ^b^		1.30 (0.56–3.04)	0.56			1.35 (0.54–3.44)	0.53	2.84 (1.2–6.83)	0.04 *	1.65 (0.71–3.85)	0.24		
WATER MEDICATED WITH ANTICOCCIDIALS														
No						Reference							Reference	
Yes					4.21 (1.38–13.83)	0.006 **							3.73 (1.10–16.3)	0.02 *
LIME														
No			Reference				Reference							
Yes			3.12 (1.41–6.94)	0.00 **			2.41 (1.01–5.67)	0.03 *						

** Differences in farm characteristics and/or control measures with *p* ≤ 0.01, * Differences in farm characteristics and/or control measures with *p* ≤ 0.05. CI: Confidence Interval. Different letters indicate significant differences.

**Table 5 pathogens-13-00048-t005:** Explanatory variables of risk for the parasitic load of *Eimeria* spp. [OPG] and the presence of *Eimeria* species using a multivariable binary logistic regression model.

Explanatory Variable	OPG		*E. acervulina*		*E. brunetti*		*E. maxima*		*E. necatrix*		*E. praecox*		*E. tenella*	
	Media	*p*-Value	OR (95% CI)	*p*-Value	OR (95% CI)	*p*-Value	OR (95% CI)	*p*-Value	OR (95% CI)	*p*-Value	OR (95% CI)	*p*-Value	OR (95% CI)	*p*-Value
DEPENDENCE														
Integrated	5459.95 ^b^	<0.01 **			Reference				Reference		Reference		Reference	
Independent	9051.97 ^a^				2.79 (1.22–6.37)	0.01 *			2.21 (1.11–4.39)	0.02 *	3.28 (1.46–7.38)	0.003 **	3.69 (1.78–7.65)	0.00 **
ALTITUDE														
Above 1300MAMSL	2582.43 ^b^	0.01			Reference								Reference	
Below 1300MAMSL	8338.06 ^a^				5.94 (1.68–1.07)	0.01 **							3.43 (1.65–8.18)	0.01 **
TYPE OF POULTRY HOUSE														
Climate-controlled poultry house							Reference							
Open-sided poultry house							3.98 (1.13–13.83)	0.03 *						
MATERIAL ROOF SOPPORT														
Metal			Reference								Reference			
Wood			2.99 (1.21–7.42)	0.02 *							3.47 (1.42–8.50)	0.01 **		
WATER SOURCE														
River or canal					Reference						Reference			
Potable water					1.64 (0.34–7.87)	0.53					0.75 (0.18–3.18)	0.69		
Well					0.30 (0.12–0.78)	0.01 *					2.59 (1.02–6.55)	0.04 *		
AGE														
More than 46 days							Reference							
Between 36 and 45 days							1.63 (0.62–4.31)	0.31						
Less than 35 days							6.93 (1.60–30.01)	0.01 **						

** Differences in farm characteristics and/or control measures with *p* ≤ 0.01, * Differences in farm characteristics and/or control measures with *p* ≤ 0.05, CI: Confidence Interval. Different letters indicate significant differences.

## Data Availability

The data presented in this study are available on request from the corresponding author.

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
