# Peer review of "Prevalence and Risk Factors of *Eimeria* spp. in Broiler Chickens from Pichincha and Santo Domingo de los Tsáchilas, Ecuador"

_pathogens, 2024, doi:10.3390/pathogens13010048_

Round 1
Reviewer 1 Report
Comments and Suggestions for Authors
Coccidia are found wherever chickens are raised, so the parasite is expected to be present in almost all broiler chicken farms. This study could provide information about the incidence of coccidia species in the major broiler chicken production areas of Ecuador.
Authors write Eimeria acervuline but the correct name is E. acervulina.
The manuscript needs improvement in the description of the protocols used for sampling the chicken farms (how many feces from each farm, the approximate weight and whether the samples were regular feces or cecal droppings) as well as the Primer’s target and sequences for both, the PCR and multiplex PCR assays.
The double sampling of 4 farms is not mentioned further in the manuscript.
The weakness of the study is the lack of information on the main Eimeria species mix found in every farm, specifying the proportion of each species in the total oocyst per gram of feces, which is different from the information shown in the manuscript where the species and frequency are reported on the basis of their global presence detected by the molecular tests in the sampled farms; however, the species and their global frequency don’t necessarily reflects the main species affecting the individual farms and the factors associated with that incidence. Adding the geographical distribution of the main Eimeria species mix found in the farms would help understand their association with factors that could determine their presence.
The total oocyst per gram of feces is reported in a wide range, from the minimum to the maximum parasitic load; however, it would be better to show in what farms that load could be associated with a potential case of coccidiosis or if the number of oocysts could be considered as “normal”, since the presence of Eimeria is almost inevitable in the broiler chicken farms. The authors can also discuss the relationship of the chickens’ age and the parasitic load, remembering the use of anticoccidial drugs and that the immune response associated with the age of the chickens affects the number of oocysts shed.
Discussion should be adjusted accordingly to the modifications.
Comments on the Quality of English Language
Please review the writing of the manuscript, there are a number of items to be corrected. For example, the authors use the term "batch" instead of flock.
Reviewer 2 Report
Comments and Suggestions for Authors
This study reports the prevalence and risk factors of Eimeria in chickens, the experiments are performed sound and valid, the analysis for the risk factors is rigorous. The manuscript is well organized and well written. However, the study also has some limitations and some issues need to be addressed.
Major issues:
1. A total of 159 samples were identified from 155 farms, the sample size is relatively limited, and concerns may rise to the real representativeness of these samples to each of the farms. I suggest more than three mixed samples taken from a farm.
2. Figure 1: the sampling farms should be labeled more specific to the map, may be dotted plotted. And the map may be better to zoomed in.
Minor issues:
1. Line 91-92: However, a total of 159 fresh feces samples was obtained on four farms?
2. Line 19, 155, 293…: “The parasitic load varied between 25 to 69.900 oocyst per gram”, 69.900 or 69,900?
3. For the statistical P value, the letter P should be capitalized and italic.
4. Line 96: “4oC”?
5. Line251-252: formatting error
Comments on the Quality of English LanguageThe quality of English is acceptable.
Reviewer 3 Report
Comments and Suggestions for Authors
The manuscript entitled: Prevalence and risk factors of Eimeria spp. in broiler chickens from Pichincha and Santo Domingo de los Tsáchilas, Ecuador is a study carried out in 155 poultry farms in a major broiler chicken production of Ecuador that aimed at determining the prevalence of Eimeria and its associated risk factors. It is a huge fieldwork that contributes to coccidiosis control not only in Ecuador but also in the region- and the world- given that there is scarce information on circulating species of Eimeria in Latin America. Importantly, the study showed an elevated -almost exclusive- use of anticoccidial drugs that are being limited or banned worldwide, which highlights their environmental impact and emphasizes the urgent need for alternative cost-effective anticoccidial strategies. However, the information regarding the operations is not well organized; for example, the production systems included in the study are not clear. Additionally, the molecular typing is not conclusive given the absence of a proper control for the species E. mitis. Thus, major as well as minor revisions should be addressed by the authors and are detailed in the file.

Some specific words and terms should be revised, they are detailed in the attached file
Round 2
Reviewer 1 Report
Comments and Suggestions for Authors
Previous issues were properly addressed. There are only a few typing corrections needed:
line 126 Table 1. 5th Column header: Amplicon instead of Aplicon
Line 227 delete "conducting breeding activities"
line 383 add "in Eimeria species" after genetic variations
line 417 delete probably
Comments on the Quality of English LanguageLine 247 replace food by feed
line 386 handling instead of handing
Reviewer 3 Report
Comments and Suggestions for Authors
This version of the manuscript has been greatly improved. I would suggest a few more issues. Given that the work did not include the identification of E. mitis this should be added to the manuscript. For example, in line 15, I suggest say: ... and the identification of six out of the seven chicken Eimeria species, the same clarification in L 176 and Table 2 (L198). Additionally, I suggest adding to the discussion that E. mitis was not analyzed in the current work, which may or may not be present, thus, its identification must be further studied to establish the true prevalence of the 7 chicken species.
In the introduction line 77, please rephrase the sentence, one suggestion could be: Given the aforementioned, the objective of the current work was to analyse the prevalence of Eimeria sp. in Ecuadorian poultry.....
I suggest replacing chicken instead of avian, since the later include also other species of birds like turkeys
L139 Does it refer to data curation? please check
Paragraph 407-418. I don't find the argument. I suggest shortening, emphasizing that oocysts are highly resistant and can persist in the ground for long terms
Please revise the italics in the whole manuscript
